# Design of Biodegradable Films Using Pecan Nut Cake Extracts for Food Packing

**DOI:** 10.3390/foods12071405

**Published:** 2023-03-26

**Authors:** Jamila dos Santos Alves, Nicholas Islongo Canabarro, Caroline Pagnossim Boeira, Pamela Thais Sousa Melo, Marcia Regina de Moura Aouada, Claudia Severo da Rosa

**Affiliations:** 1Programa de Pós-Graduação em Ciência e Tecnologia dos Alimentos, Universidade Federal de Santa Maria, Santa Maria 97105-900, RS, Brazil; 2Bioenergy and Catalysis Laboratory (LBK), Paul Scherrer Institute, 5232 Villigen, Switzerland; 3Hybrid Composites and Nanocomposites Group, Department of Physics and Chemistry, Universidade Estadual Paulista (UNESP), Ilha Solteira 15385-000, SP, Brazil

**Keywords:** pecan nut cake, phenolic compounds, biodegradable films, biopolymers

## Abstract

The excessive consumption of plastic packaging and its consequent disposal and accumulation in the environment have aroused the interest of researchers in developing packaging that can cause less harm to nature. In this sense, this article presents research on the addition of antioxidant extracts from pecan nut cake in biodegradable packaging made with a polymeric mixture of gelatin and corn starch. The films produced were characterized by scanning electron microscopy (SEM), Fourier transform infrared (FTIR) spectroscopy, thickness, mechanical properties, water vapor permeability (WVP), solubility, water contact angle, optical properties, in vitro bioactive activity, and biodegradability. A higher concentration of total phenolic compounds (101.61 mg GAE/g) was found for the condition where alcohol content and extraction time were 65% and 20 min, respectively. Pecan nut cake (PNC( extracts did not influence the film’s tensile strength, and elongation at break was tightly increased by adding 10–20% extracts. The film’s characterization pointed to more than 67% solubility, and adding PNC extract implied more hydrophilic surfaces (contact angles lower than 65°). Furthermore, the film opacity showed a linear relation with PNC extract concentration, and a higher luminosity (L*) was observed for the film without extract. Furthermore, the antioxidant activity of the films was enhanced with the addition of PNC extracts, and complete biodegradation was observed until the ninth day. Therefore, biodegradable films prepared from a mixture of gelatin starch and enriched with PNC extracts showed excellent mechanical properties and potential as carriers of antioxidant compounds, allowing us to propose their use as active packing.

## 1. Introduction

Attributable to their positive effects on health [1], flavor, and pleasant texture [2], the pecan nut (*Carya illinoinensis* (Wangenh.) K. Koch) is one of the largest nuts consumed worldwide [3]. Generally, pecan nuts are consumed in their fresh form and used to prepare bakery and confectionery products [4]. Lately, knowledge about the benefits of pecan oil has drawn consumers’ attention [1].

Pecan nut oil can be obtained by an extraction process that uses organic solvents, pressurized fluids, and mechanical cold pressing [5]. The oil extraction process produces a by-product known as pecan nut cake (PNC), which possesses valuable and interesting industrial compounds in its composition, such as bioactive compounds. Phenolic compounds are found in large amounts in PNC extracts, and the main compounds are catechin, epigallocatechin, and epicatechin [6]. Therefore, the development of bioactive extracts from PNC becomes a sustainable way to reuse and add value to a byproduct that is usually destined for animal feed [7]. 

Recently, concerns about the environmental impacts of the increased use and disposal of petrol-based plastic packaging in nature have claimed the scientific community’s attention [8]. For this reason, developing polymer-based biodegradable films has been considered a reliable way to replace petrol-based packaging [9]. Proteins and polysaccharides are the main biopolymers used to elaborate films for biodegradable packaging [10]. Among them, corn starch and gelatin stand out, and the blending of both can result in films with enhanced properties. Since 1980, starch has been used to produce biodegradable matrices, being one of the most abundant, biodegradable, and renewable natural biopolymers derived from various plant sources [11]. Starch is widely used due to its excellent film-forming properties, neutral organoleptic properties, and low cost [12]. Gelatin is a biopolymer with great potential for use in packaging production due to its properties, such as good film-forming capacity, biocompatibility, non-toxicity, abundance, transparency, and potential use as a carrier of various compounds and additives [13].

In order to enhance the biological properties of the films, as well as their physical and functional properties, different antioxidant and antimicrobial components have been added to the film-forming solution [14]. Additionally, bioactive compounds in biodegradable films interact with the packaged product [15]. According to Silva-Weiss et al. [16], these compounds protect foods against oxidation, microorganisms’ proliferation, enzymatic browning, and loss of vitamins, enhancing the functional properties of films.

Given the above, the PNC is a by-product that allows the extraction of bioactive compounds with commercial interest. Therefore, this work aimed to extract the phenolic compounds from PNC and apply the extract with the highest concentration in films prepared with a mixture of gelatin and starch. The films were evaluated on their antioxidant characteristics, solubility, angle of contact with water, morphological properties, biodegradability, and color.

## 2. Materials and Methods

### 2.1. Materials and Chemicals

The PNC was donated by Nozes Pitol Grupo Alimentos (Anta Gorda, RS, Brazil). It was stored in plastic packaging under vacuum at −18 °C until use. The PNC was received partially defatted, which means it was collected just after the oil extraction at the plant. Materials used for the production of the films were gelatin type A (SM Empreendimentos Farmacêuticos LTDA, São Paulo, SP, Brazil), native corn starch supplied by Maizena^®^ (Brazil), and glycerol PA (Neon Comercial de Reagentes Analíticos, Suzano, Brazil).

### 2.2. Extraction of Samples 

The extracts were obtained through the methodology described by Sarkis et al. [8]. Briefly, 5 g of PNC were added to 50 mL of hydroethanolic solution. The mixture was maintained under constant agitation (Marconi MA039, Piracicaba, SP, Brazil) and a temperature of 25 °C, controlled using an ultra-thermostatic bath (Marconi, MA-184, Piracicaba, SP, Brazil). The influence of the extraction parameters, ethanol concentration (%), and extraction time (min) on the total phenolic content (TPC) was investigated through a factorial experimental design 2² with triplicates at the central point (Table 1). The liquid portion was separated from the solid ones by filtration (filter paper, J. Prolab, São José dos Pinhais, SP, Brazil). The extract purification (remove ethanol) was performed using a rota evaporator at 45 °C (R-300, Büchi Brazil Ltda, Valinhos, SP, Brazil).

#### Phenolic Compounds Evaluation

The total phenolic content was determined by the Folin–Ciocalteu method described by Roesler et al. [17], and the results were expressed in milligrams of gallic acid per gram of partially defatted PNC (^1^ mg GAE/g).

### 2.3. Film Preparation

The films containing 5% gelatin and 2% starch were prepared by a casting technique, according to Malherbi et al. [18]. The control film was prepared as follows: 5 g of gelatin was hydrated with 100 mL for 1 h at 25 ± 2 °C. Later, to ensure the solubilization of gelatin, the solution was taken to an ultra-thermostatic bath (Marconi, MA-184, Piracicaba, Brazil) at 55 °C for 10 min. Glycerol (10 wt.% concerning gelatin) was added to the solution under constant agitation (magnetic agitation). Finally, the starch solution was prepared by dissolving 2 g of corn starch into 100 mL of distilled water and glycerol (10 wt.% in regards to gelatin) under manual agitation at 80 ± 2 °C for 15 min. Then, the gelatin and starch solutions were mixed in a proportion of 1:1 under constant agitation (magnetic agitation for 1 min at 25 ± 2 °C). For the preparation of films containing PNC extracts (“active films”), 10% and 20% of the extract were added, depending on the amount of distilled water used in the preparation of the gelatin and starch solution. The films were poured into a polyester holder, and the thickness of the film-forming layers were adjusted with a ruler. Afterwards, they were dried at 30 ± 2 °C for 24 h in a forced convection oven. The concentration of PNC extracts was chosen based on preliminary results obtained in our group (not presented). Briefly, films in which more than 20% of extracts were tested have not presented improvements in antioxidant activity and have supported the concentration range adopted in this study. 

### 2.4. Characterization of Films

#### 2.4.1. Scanning Electron Microscopy (SEM)

The EVO LS15 scanning electron microscope (Carl Zeiss, München, Germany) was utilized to examine the fracture surfaces and cross-sectional structures of the films. The samples were initially affixed to a bronze stump and coated with carbon tape before being sprayed with gold and aspirated using the Quorum Q150TE Sputter Coater (Laughton, East Sussex, England) for 20 min. The visualization analysis was conducted with magnification levels ranging from 1000 to 10,000× and an acceleration voltage of 5–10 kV was employed.

#### 2.4.2. FTIR

The samples were macerated with potassium bromide (KBr) and then pressed to obtain pellets. The FTIR spectra of all films were obtained using an FTIR spectrometer (Nicolet Instrument Corporation, Waltham, MA, USA) in the spectral region of –. The spectra were collected with a resolution of 4 cm^−1^, and a total of 128 consecutive scans were recorded.

#### 2.4.3. Thickness

A digital micrometer measured the film’s thickness in triplicate (n° 7326, Mitutoyo Corp., Sakado, Japan). The mean values were also used to obtain the mechanical and optical properties of the films. 

#### 2.4.4. Mechanical Properties

The film samples (100 mm long and 13 mm wide) were prepared according to ASTM D882/97 [19] and characterized by their mechanical properties, tension, and elongation at break. Measurements were conducted on a universal testing machine (model 3369, Instron Corp., Norfolk County, MA, USA), operating with a 50 N load cell with a traction speed of 10 mm.min^−1^. All assays were performed at 30 ± 2 °C and a relative humidity of 50 ± 2%. The tension at break (σ) (Equation (1)) and the elongation at break (ε) (Equation (2)) were obtained as follows:(1)$$\sigma \left(\mathrm{MPa}\right)=\frac{F}{S}$$
(2)$$\varepsilon \;\left(\%\right)=\left(\frac{\Delta L}{L_{0}}\right)\times 100$$
where F is the breaking force exerted, S is the cross-sectional film area, ∆L is the film elongation at the end of the procedure, and L_0_ is the initial film elongation.

#### 2.4.5. Water Vapor Permeability (WVP)

The water vapor permeability of the films was determined by the gravimetric method, according to the ASTM E96-80 [20] method and modified by Mchugh et al. [21]. The film samples were cut into circles and fixed in standardized cells containing 6 mL of distilled water. The cells were maintained at 25 ± 2 °C in an oven containing silica gel to control the film’s moisture. The periodic weighing of cells determined the amount of water that permeated through the film for 24 h, and WVP was defined according to Equation (3):(3)$$WVP=\frac{g.e}{A.t.\Delta P}$$
where WVP is the water vapor permeability (g·m/m^2^·s·Pa); g is the mass change of the cell; t is the time (hours); A is the cross-sectional area (m^2^); e is the mean thickness of the films (mm); ∆P is the pressure drop in the film (kPa, 30 °C); the term “g/t” was estimated by linear regression. 

#### 2.4.6. Solubility

The solubility of films was determined according to the method described by Peña et al. [22]. Briefly, to determine the sample dry mass (m_i_), discs with a diameter of 2 cm were prepared and dried in an oven at 100 ± 2 °C for 24 h. The determination of the water content in the films (non-solubilized dry mass, m_f_) was carried out as follows: the samples were immersed in 50 mL of distilled water and kept under constant agitation (70 RPM) and temperature (25 ± 2 °C) for 24 h. Then, the samples were placed in an oven at 100 ± 2 °C for 24 h. Finally, the samples were withdrawn from the oven and taken to a desiccator to avoid humidity absorption until they reached room temperature. The solubility percentage (S%) was obtained according to Equation (4):(4)$$S=\frac{\left(m_{i}-m_{f}\right)}{m_{i}}\times 100$$
where m_i_ is sample dry mass and m_f_ is non-solubilized dry mass.

#### 2.4.7. Water Contact Angle

Contact angle measurements were performed using a contact angle meter (Biolin Scientific, Optical Tensiometer Theta Lite, Stockholm, Sweden). A drop (5–9 μL) of Mili-Q water was deposited on the film surface, and the contact angles were determined after 10 s [23]. The contact angle value was calculated by averaging the angles at the right and left ends of the drop.

#### 2.4.8. Optical Properties

Color measurements were realized based on the CIELab color scale. The parameters L* (luminosity), a* (red to green), and b* (yellow to blue) were measured randomly at five distinct points through a spectrophotometer (Model CM-700d, Konica Minolta, Japan). The values of a* and b* were used to calculate h° (hue angle) (Equation (5)) and C* (color intensity) (Equation (6)).
(5)$$h{}^{\circ}=tan^{-1}\;\left(\frac{b^{*}}{a^{*}}\right)$$
(6)$$C*=\sqrt{{\left(a^{*}\right)}^{2}+{\left(b^{*}\right)}^{2}}$$

The opacity (Equation (7)) was estimated according to the procedure described by Souza et al. [24]. Briefly, the films were cut into rectangles, and their absorbances were measured at 600 nm on a spectrophotometer UV/VIS (Minolta Sensing Inc., Konica, Chiyoda-ku, Japan). An empty cell was used as a reference, and the opacity was obtained by the ratio between absorbance and film thickness (mm).
(7)$$Opacity\;\left(mm\right)=\frac{Absorbance_{600nm}}{Film\;thickness\;\left(mm\right)}$$

#### 2.4.9. In Vitro Bioactive Activity of Films

Antioxidant activity (AA%) of the films was evaluated by the DPPH (2,2-diphenyl-1-picrylhydrazyl) free radical scavenging activity protocol, described by Shahbazi et al. [25]. Preliminary tests were carried out to define the ideal concentration of films. The films (50 mg) were solubilized in 10 mL of a 65% hydroethanolic solution under magnetic stirring (Biomixer model 78HW-1) at 40 °C and centrifuged at 8000 g for 10 min. The supernatant was used for the analysis of antioxidant activity. A spectrophotometer measured the sample’s absorbance at 517 nm (Model CM-700d, Konica Minolta, Japan). The results were expressed as a percentage (%) of inhibition of DPPH radical oxidation according to Equation (8):(8)$$AA\%=\frac{\left({Abs}_{DPPH}-{Abs}_{extract}\right)}{{Abs}_{DPPH}}\times 100$$
where Abs_DPPH_ and Abs_extract_ are the absorbance of the DPPH solution and sample, respectively.

#### 2.4.10. Biodegradability

The qualitative test of the film’s biodegradability was carried out as proposed by Boeira et al. [26]. The vegetable compost (soil) was poured into a plastic tray (15 × 30 × 10 cm) up to a height of about 7 cm. Film samples (2 cm × 3 cm) were buried in the soil at a depth of approximately 2 cm. The plastic trays were kept at room temperature (25 °C), and water was sprayed twice a day to maintain the humidity of the medium. At different times (the first, third, sixth, and ninth days), the samples were carefully collected through photographic recording and visual assessment of biodegradation.

### 2.5. Statistical Analysis 

A 2² experimental design (central composite design—CCD) with a triplicate in the central point was performed to investigate the effects of alcohol content and extraction time on the antioxidant activity of PNC extracts. The analysis of variance (ANOVA) was applied to the experimental data at a significance level of 5% using the software Statistica 7.0 (Statsoft Inc., Tulsa, OK, USA). The Tukey test was applied to evaluate the statistical differences among the samples.

## 3. Results and Discussion

### 3.1. Total Phenolic Compounds of PNC Extracts 

The total phenolic compound (TPC) content of PNC extracts obtained under different extraction conditions is presented in Table 1, and the results showed significant differences. 

The higher TPC content (101.6 mg GAE/g) was observed when lower alcohol content and extraction time were employed (T1—alcohol 65% and 20 min), while the inferior extraction performance was achieved for assay T4 (alcohol 95% and 40 min). The TPC content obtained in this work is higher than that reported by Maciel et al. [27], where they evaluated the PNC extracts obtained by three different solvents (water, acetic acid, and ethanol) and reached values of 1.7 to 27.4 mg GAE/g. It is worth noting that the results of the Folin–Ciocalteu method are difficult to compare, as some factors such as structural variation of phenolics within and between species, presence of interfering metabolites, and varying extraction and reaction conditions can either under- or overestimate the obtained results [28]. Therefore, a comparison between different works must be made considering these aspects, and a deeper (and fairer) discussion was made only between the results obtained in this work.

Table 2 presents the effects of the variables, alcohol content and extraction time, on the TPC content. As can be seen, the impact on TPC content is harmful for both variables (*p*-value < 0.01), which means the increases in alcohol concentration and extraction time reduce the TPC content. Therefore, regarding alcohol content, it is possible to infer that the alcoholic solution of 65% has high polarity and better physical properties than the others, which can enhance the concentration of TPC from the PNC extracts. The polarity of the alcohol is inversely proportional to the chain size, meaning that the ethanol has a suitable polarity to bind to the phenolic compounds in the pecan pie and extract them [29]. Furthermore, ethanol solutions have presented interesting results in TPC extraction. The work performed by Dewi et al. [30] showed that solvent selection is a critical parameter in enhancing the TPC in cacao pod husk. The authors investigated the impact of different solvents (methanol, ethanol, 1-propanol, 1-butanol, 1-pentanol, deionized water, 50% (*v*/*v*) aqueous methanol, and 50% (*v*/*v*) aqueous ethanol) on TPC using the Hansen solubility parameter (HSP) prediction to define the miscibility behavior of solvents with gallic acid as a standard phenolic compound. The HSP method estimates the distance between two molecules (called Ra), which measures the similarity between the molecules and describes their compatibility. The lower the Ra, the higher the compatibility between the molecules. The results showed lower Ra values for 50% (*v*/*v*) aqueous ethanol and gallic acid, which means this solvent presented the best performance for extracting phenolic compounds.

According to Kiralan et al. [31], pecan nut oil contains the following phenolics: gallic acid, protocatechuic, catechin, ferulic, sinapic, naringenin, chlorogenic, and luteolin. Therefore, the solubility behavior found in this work follows the same trend as reported by Dewi et al. [30], which means increasing the ethanol concentration increases the distance between the phenolic compounds and solvent. This behavior does not favor phenolic compound extraction. Extraction times higher than 20 min resulted in lower TPC content, probably due to medium saturation and TPC degradation by oxidation. 

### 3.2. Films Characterisation

#### 3.2.1. Scanning Electron Microscopy (SEM)

The addition of PNC extracts by the films (GA-10% and GA-20%) resulted in a more irregular and less smooth fracture surface than the film without extract, as shown in Figure 1. On the other hand, the less homogeneous structure observed for the films GA-10% and GA-20% may be explained by the presence of distinct chemical compounds that belong to the PNC extracts. Despite this, no phase separation was observed on the surface and cross-sections, evidencing the excellent miscibility of the compounds in the films. Zhang et al. [32] reported that the addition of extracts forms hydrogen bonds, producing more compact and denser starch networks. The cross-sections of the films presented multilayers, which are typical of starch films [33].

#### 3.2.2. Fourier Transform Infrared Spectroscopy (FTIR)

FTIR spectroscopy was performed to investigate the influence of PNC extracts on the gelatin–starch film structure, and the results are presented in Figure 2. In general, no relevant changes were observed when PNC extracts were incorporated into the films. The spectrum of GA film showed characteristic bands at approximately 3100–3600 cm^−1^ (amide-A, N-H stretching; O-H stretching), 2850–3100 cm^−1^ (C-H stretching, -OH stretching), 1637 cm^−1^ (Amide I, C=O stretching), 1385 cm^−1^ (aromatic hydroxyl groups), 1158 cm^−1^ (C-O-C stretching), and 1035 cm^−1^ (-OH groups of glycerol coupled to -CH2 of amino acid residues of gelatin molecules) [34,35,36]. The spectra of GA-10% and GA-20% showed similar peaks and characteristic bands attributed to GA. However, the peak amplitude for the wavenumber at 2850–3100 cm^−1^ increased as the PNC extract concentration increased. The spectral differences are related to the presence of aliphatic and unsaturated hydrocarbons related to terpenoid components, which were enhanced due to PNC extracts’ higher concentrations incorporated into the film [36]. The opposite occurred for the wavenumber at 1035 cm^−1^, where the peak amplitude decreased as PNC extract concentration increased. In this case, PNC extracts might replace the glycerol (plasticizer) presented in gelatin–starch films, which can affect the films’ mechanical properties [34].

#### 3.2.3. Thickness

The thickness of the films was not significantly affected by the incorporation of PNC extracts, as shown in Table 3, where values ranged from 0.036 to 0.039 mm. This result can be explained by the presence of hydroxyl groups of polyphenols present in PNC, which formed intermolecular hydrogen bonds with the amino/hydroxyl groups of gelatin and were thus uniformly distributed in the space between the film matrix [37]. Generally, the commercial plastic film thicknesses of low-density polyethene (LDPE) and polypropylene (PP) () are around 0.015–0.25 mm and 0.012–0.125 mm, respectively [38], comprising the thickness range of the films prepared in this study. Nur Amila Najwa et al. [39] found higher values than those obtained in this study for tapioca starch/gelatin films incorporated with *Garcinia atroviridis* extract (0.094 to 0.101 mm). They reported that the incorporation of the extract had no effect on the thickness of the films.

#### 3.2.4. Mechanical Properties

The results for the mechanical properties of the films, such as tensile strength (TS) and elongation at break (EB), are presented in Table 3, where TS represents the maximum strength a film can resist and EB measures the film flexibility [40]. The films shall possess appropriate mechanical properties to avoid any changes in food integrity during handling and storage [41].

The results stressed the possibility of adding nutraceuticals to gelatin/starch films without significant changes in the films’ mechanical properties. Although some authors have reported that adding a bioactive agent decreased the film’s TS significantly [26,34,35], the concentration of PNC extracts incorporated into gelatin/starch films does not influence the film’s TS. However, the addition of the bioactive agent increased the EB of the films. For GA film, the EB value was 3.53%, while for GA-10% and GA-20%, it was 4.42 and 4.18%, respectively. Although there was no statistical difference between the GA and GA-20% films, there was a significant difference when comparing the GA and GA-10% films. This means that adding 10–20% (% by weight) of PNC extracts improves film strength without relevant structural changes (see Section 3.2.2). Hosseini et al. [42] showed that the inclusion of essential oils reduces the intrinsic stiffness of gelatin/chitosan films. Our study also corroborates with that of Xu et al. [43], who reported that adding grape extract to corn starch nanocomposite films raises EB values. According to the authors, this behavior can occur due to the extracts’ action as plasticizers. Therefore, adding 10–20% (%wt) of PNC extracts in gelatin/starch films positively affects the film’s mechanical properties, maintaining similar values of TS and slightly enhancing the EB compared to the control sample (GA).

#### 3.2.5. Water Vapor Permeability (WVP)

As shown in Table 4, the PNC extracts have no influence on the WVP of the films since the results do not show statistical differences. Although adding PNC extracts implied an increase in the films’ hydrophilicity (see Section 3.2.7) due to the presence of hydroxyl groups (related to phenolic compounds present in the extracts’ composition), the amounts of PNC extracts were not enough to change the WVP. Even though the results do not show a statistical difference, we can observe that the GA-20% film had the lowest WVP values. One possible explanation is that the decrease in WVP is related to the polyphenols from the PNC extracts, which may be interacting with the polysaccharide-based film matrix through hydrogen bonding and electrostatic effects [37]. Similar results were found by Crizel and Costa [44], who developed gelatin films with blueberry extracts. The author showed that blueberry extracts do not affect the WVP; however, the films presented higher values of WVP and thickness than those obtained in this work.

On the other hand, Albertos et al. [45] produced gelatin films that incorporated olive leaf extracts and reached WVP values worse than those obtained in this work (1.83 to 4.67 × 10^–10^ g·m/m^2^·s·Pa). According to Mali et al. [46], the main function of the films is to prevent moisture transfer between food and package or between two components within a heterogeneous food product. Therefore, films utilized in food should have WVP as low as possible.

#### 3.2.6. Solubility

Solubility of films varied from 67.28 ± 3.57% (GA-10%) to 69.38 ± 3.35% (GA-20%) (Table 4) and statistically has no expressive differences since the fluctuations in values are within the error of measure. The high solubility found may be due to the hygroscopic nature of starch and gelatin, which rapidly disintegrate in water [47]. The study conducted by Bitencourt et al. [15] investigated the solubility of gelatin films with different concentrations of turmeric extracts and achieved lower values than those obtained in this work (35.4 and 40.0%). According to Fakhouri et al. [48], solubility in water is an important parameter to assess since it can help with the decision of the suitable film used for a specific food. For instance, film solubilization is beneficial in semi-finished products that require cooking in their preparation.

#### 3.2.7. Water Contact Angle

The contact angle measures the film surfaces’ hydrophilicity, or the hydrophilicity degree [40]. Low water contact angles (θ < 65°) indicate a hydrophilic surface, while high contact angle values (θ > 65°) correspond to a hydrophobic surface [49,50]. As can be seen in Table 4, the contact angle (CA) for the GA film has hydrophobic characteristics (CA 70.1 ± 1). The addition of PNC extract caused a significant decrease in CA values, indicating the hydrophilic character of the surface of the films (GA-10% and GA-20% have contact angles of 64.9 ± 2 and 59.9 ± 5.9, respectively). The same statement was observed by Kaya et al. [51], who elaborated on chitosan films with *Berberis crataegina* fruit extracts and reported contact angle decrease due to the hydrophilicity of the extracts. Thus, the incorporation of PNC extracts increased the tendency of water droplets to spread on the surface of the film, reducing the water contact angle [52].

#### 3.2.8. Optical Properties

Opacity is an important factor in controlling the amount of light that passes through food [47]. Table 5 presents the opacity values of the films and shows the influence of PNC extract addition on the results. As can be observed, there is a slight dependence between opacity and PNC extract concentration, as a small increase in opacity is observed when the concentration of TPC extracts is increased. In ascending order, the opacity values were 1.32 ± 0.03, 1.81 ± 0.13, and 1.96 ± 0.06 for the film GA, GA-10%, and GA-20%, respectively. The work performed by Kan et al. [53], revealed that the opacity of films elaborated with a gelatin–chitosan mixture increased when *Crataegus pinnatifida* extracts were added to the films, which corroborates the results achieved in this study. According to the authors, the extract’s addition alters the film’s optical characteristics and interferes with the absorption wavelength. It is worth noting that the same behavior was found in this study. Li et al. [54] reported that the -OH group present in phenolic compounds reduces light transmission, which is consistent with the results obtained in this study due to the presence of these compounds in PNC extracts (see Section 3.1).

The color of films is one of the most critical parameters regarding the overall appearance of food products and consumer acceptance [55]. The addition of PNC extracts influences the film colors, as shown in Table 5. The luminosity (L*) was higher in the film with no added extract (GA) and differed statistically from the films with the added extract. Chroma (C*) was higher in GA 20% (6.03 ± 0.21) and GA-10% (5.95 ± 0.05), indicating that the staining of these films is more intense compared to G.A. (4.94± 0.14). The hue angle (h°) of G.A. showed the lowest values (287.35 ± 0.17), and increased significantly as the concentration of extract in the film was raised (GA-10%—292.56 ± 0.26; GA-20%—298.33 ± 0.73). According to Souza et al. [24], color changes in the films elaborated with the addition of extracts may be due to the extract itself or due to the chemical interaction between the extract and film polymer, in which the extract properties and its concentration may influence.

#### 3.2.9. In Vitro Bioactive Activity of Films

The antioxidant activities of the films are presented in Figure 3. The use of PNC extracts shows to have a positive effect on the antioxidant activity of the films, as the values were 3.32 ±1.25 (G.A.), 46.23 ± 0.69 (GA-10%), and 84.31 ± 1.29 (GA-20%), respectively. Silva et al. [56] added loquat leaf extracts (*Eriobotrya japônica*) in films elaborated with a mixture of banana peel and starch.

The authors observed that the antioxidant activity of the films was enhanced with the addition of loquat leaf extracts, agreeing with the achievements in this study. De Carli et al. [57] developed chitosan-based biodegradable active films enriched with propolis extracts (PS). They found that the addition of propolis resulted in a significant improvement in the antioxidant activity of the films. The DPPH radical scavenging capacity increased to 49.8%, 94.0%, and 94.5% for films with 5% PS, 10% PS, and 20% PS, respectively. According to the authors, the inclusion of natural extracts in films increases the stability against oxidation of packaged food products due to their antioxidant capacity [58,59].

#### 3.2.10. Biodegradability

The films’ biodegradability as a function of the days they were left in contact with soil is shown in Figure 4.

The biological decomposition started on the third day and rose until the ninth day, when the total film decomposition occurred. It is worth noticing a tight difference between the biodegradability rate of the G.A. sample and those in which PNC extracts were used. As shown in Figure 4, the biodegradability rate increased with the decrease in PNC extract concentration in the films, indicating that the bioactive compounds have negatively affected the microorganisms’ activity in the soil. The results on biodegradability found by Piñeros-Hernández et al. [60] showed that the integrity of the films was affected when rosemary (*Salvia rosmarinus*) was incorporated into them. Therefore, a higher resistance in film biodegradability is due to the bioactive compounds’ activity in the extracts. A package is considered biodegradable when 90–95% of its structure is decomposed within 6 months and the process ends [61]. Therefore, the result achieved in this work demonstrates that the films obtained can be considered biodegradable.

## 4. Conclusions

Biodegradable films developed with a mixture of biopolymers (gelatin–starch) added with PNC extracts have been successfully elaborated. The addition of 10 and 20% extract tightly changed the films’ surfaces but did not change the molecular interaction in the formed films. PNC extracts did not impact TS and smoothly increased the EB in films, preserving the mechanical properties presented by the films without PNC extracts. Adding pecan extract implied more hydrophilic surfaces (contact angles lower than 65°) and higher film opacity than the control one (GA). Furthermore, incorporating extracts into the films significantly increased the antioxidant activity, mainly due to the presence of phenolic compounds in PNC extracts, and did not affect the film biodegradation that occurred entirely on the 9th day. Therefore, biodegradable films made with a starch–gelatin mixture enriched with PNC extract present excellent mechanical properties and are potential carriers of antioxidant compounds, allowing this edible film to be used as active packaging in foods. Films based on starch–gelatin incorporated with PNC extracts are ecologically correct and follow a worldwide trend of reducing oil-based plastic packaging.

## Figures and Tables

**Figure 1 foods-12-01405-f001:**
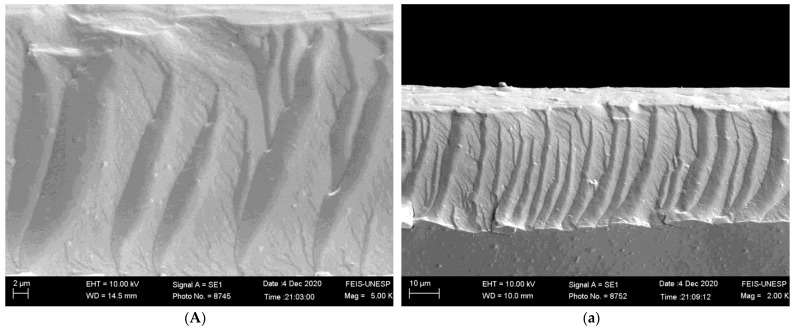
Scanning Electron Microscopy (SEM) images of gelatin and corn starch films prepared with different concentrations of PNC extract. (**A**,**a**) GA: a mixture of gelatin and starch; (**B**,**b**) GA-10%: a mixture of gelatin and starch with 10% PNC extract; (**C**,**c**) GA-20%: a mixture of gelatin and starch with 20% PNC extract.

**Figure 2 foods-12-01405-f002:**
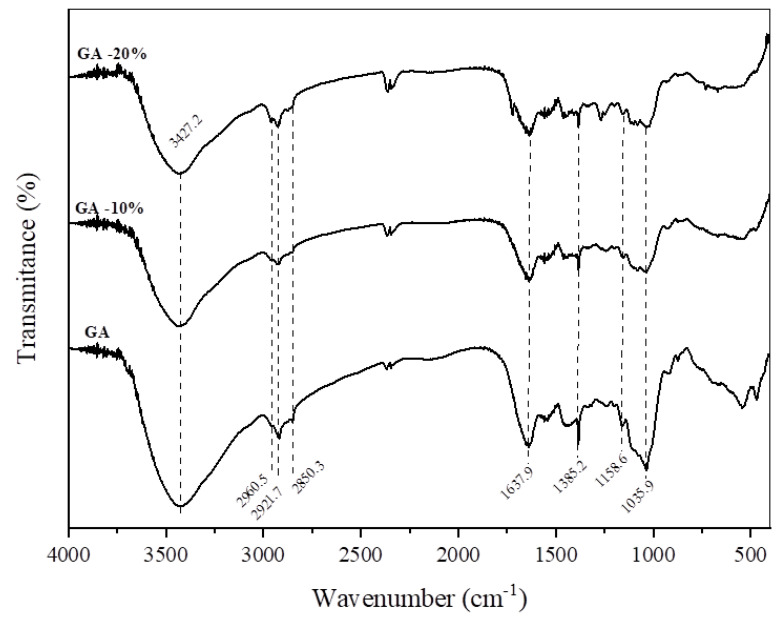
FTIR spectra of gelatin/starch films. (GA: film with no addition of PNC extracts; GA-10%: film with the addition of 10%wt of PNC extracts; GA-20%: film with the addition of 20%wt of PNC extracts).

**Figure 3 foods-12-01405-f003:**
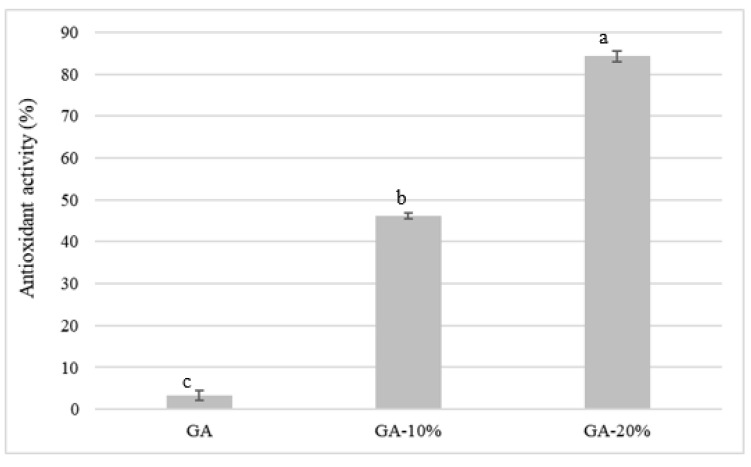
Antioxidant activity of gelatin–starch blend films added to PNC extract (*n* = 3). GA: a mixture of gelatin and starch; GA-10%: a mixture of gelatin and starch with 10% PNC extract; GA-20%: a mixture of gelatin and starch with 20% PNC extract. Different letters indicate a significant difference (*p* < 0.05) by Tukey’s test.

**Figure 4 foods-12-01405-f004:**
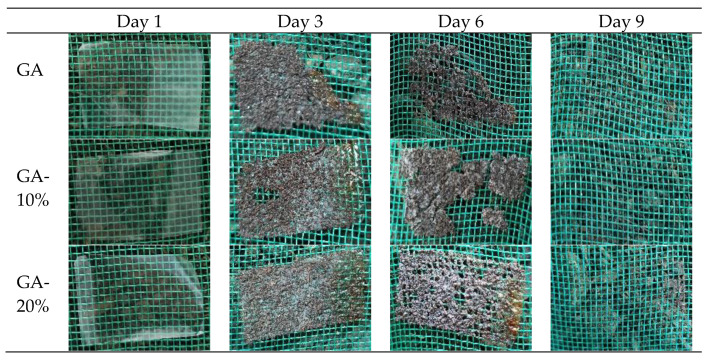
Biodegradability of gelatin–starch blend films added with PNC extracts. GA: a mixture of gelatin and starch; GA-10%: a mixture of gelatin and starch with 10% PNC extract; GA-20%: a mixture of gelatin and starch with 20% PNC extract.

**Table 1 foods-12-01405-t001:** Effects of solvent content and time extraction on the total phenolic compound (TPC) concentration of PNC extracts.

Assay	Alcohol Content (%)	Time (min)	TPC (mg EAG/g of Sample)
T1	65 (−1)	20 (−1)	101.6
T2	65 (−1)	40 (+1)	50.5
T3	95 (+1)	20 (−1)	54.5
T4	95 (+1)	40 (+1)	36.9
T5 *	80 (0)	30 (0)	40.9
T6 *	80 (0)	30 (0)	42.3
T7 *	80 (0)	30 (0)	39.5

* T5, T6, and T7 correspond to the central point.

**Table 2 foods-12-01405-t002:** Effects of alcohol content and time on the content of total phenolic compounds of PNC extracts.

	Effects	Standard Deviation	*p*-Value
Total phenolic compounds			
Interception	52.3257	0.3156	<0.01 *
(1) Alcohol content	−30.3450	0.8350	<0.01 *
(2) Time	−34.3750	0.8350	<0.01 *
1 × 2	16.7350	0.8350	0.2235

1 × 2 = interaction between Alcohol content and time. * There was a significant effect considering a significance of 95%

**Table 3 foods-12-01405-t003:** Thickness, tensile strength (TS), and elongation at break (EB) of films made of a mixture of gelatin and starch with added PNC extract (*n* = 3).

Sample	Thickness (mm)	TS (MPa)	EB (%)
GA	0.039 ± <0.01 ^a^	62.99 ± 7.67 ^a^	3.53 ± 0.62 ^b^
GA-10%	0.036 ± <0.01 ^a^	58.82 ± 2.53 ^a^	4.42 ± 0.56 ^a^
GA-20%	0.039 ± <0.01 ^a^	62.87± 4.77 ^a^	4.18 ± 0.68 ^ab^

When values are expressed as mean ± standard deviation, different letters in the same column indicate a significant difference (*p* < 0.05) by the Tukey test. GA: a mixture of gelatin and starch; GA-10%: a mixture of gelatin and starch with 10% PNC extract; GA-20%: a mixture of gelatin and starch with 20% PNC extract.

**Table 4 foods-12-01405-t004:** Water vapor permeability (WVP), solubility, and contact angle of gelatin–starch blend films with added PNC extract (*n* = 3).

Sample	WVP (g·m/m^2^·s·Pa)	Solubility (%)	Contact Angle (θ °)
GA	1.03 × 10^−10^ ± <0.01 ^a^	69.10 ± 1.49 ^a^	70.11 ± 0.98 ^a^
GA-10%	1.10 × 10^−10^ ± <0.01 ^a^	67.28 ± 3.57 ^a^	64.90± 2.08 ^ab^
GA-20%	9.44 × 10^−11^ ± <0.01 ^a^	69.38 ± 3.35 ^a^	59.95 ± 5.90 ^b^

When values are expressed as mean ± standard deviation, different letters in the same column indicate a significant difference (*p* < 0.05) by the Tukey test. GA: a mixture of gelatin and starch; GA-10%: a mixture of gelatin and starch with 10% PNC extract; GA-20%: a mixture of gelatin and starch with 20% PNC extract.

**Table 5 foods-12-01405-t005:** Optical properties of gelatin–starch blend films with the addition of PNC extract (*n* = 3).

Sample	Opacity (mm)	L*	C*	h°
GA	1.32 ± 0.03 ^b^	90.59 ± 0.27 ^a^	6.03 ± 0.21 ^a^	287.35 ± 0.17 ^c^
GA-10%	1.81 ± 0.13 ^a^	89.70 ± 0.23 ^b^	5.95 ± 0.05 ^a^	292.56 ± 0.26 ^b^
GA-20%	1.96 ± 0.06 ^a^	88.79 ± 0.12 ^c^	4.94± 0.14 ^b^	298.33 ± 0.73 ^a^

When values are expressed as mean ± standard deviation, different letters in the same column indicate a significant difference (*p* < 0.05) by the Tukey test. L* (luminosity); C* (color intensity); h° (hue angle). GA: a mixture of gelatin and starch; GA-10%: a mixture of gelatin and starch with 10% PNC extract; GA-20%: a mixture of gelatin and starch with 20% PNC extract.

## Data Availability

All data generated or analyzed during this study are included in the submitted version of the manuscript. Data is contained within the article.

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
