# Peer review of "Design of Biodegradable Films Using Pecan Nut Cake Extracts for Food Packing"

_foods, 2023, doi:10.3390/foods12071405_

Round 1

Reviewer 1 Report

Dear Authors,

This paper deals with the preparation and characterization of biodegradable films using pecan nut cake extracts for food packing. There are some points that need attention.

Abstract is informative but must include all the characterization techniques used in films

Introduction section. It must be improved, giving examples of starch - gelatin film's previous application. Also, point to the properties of polyphenolic compounds previously isolated from pecan and why the proposed method is a new study?. Besides, highlight the reasons why these mixtures can be interesting for use in food packaging (e.g. antioxidant, antimicrobial, etc.

Material section. 

IN SEM analysis, the samples were sputtered with gold?

In FTIR analysis, how many scans per sample were used? and wich resolution were used for recording spectra?.

Line 119..elongation at break.

Line 121 ... mmm-1(upper case)

Line 135 ...m2 (upper case)

Line 169...defne?

Results section.

Line 196-197..significant differences (add p-value).

Line 204 (needs references)

Table 1 is missed and must be included, otherwise, it is difficult to continue evaluating your work.

Identification of polyphenolic components is suggested by HPLC. Explain why it was necessary to perform this new study?

Line 306... some authors also point out that extract addition increase WVP, providing a plausible explanation of your results.

why WVP decrease when the contact angle decreases also? please explain this.

Figure 1..SEM pictures must be enlarged and a side view of the films must be provided to prove there is a film.

Line 259-and 263...cm-1 (cap letter).

Figure 2. FTIR spectra must be provided in a larger version and using black lines for all spectra . The way they are presented do not provide any information.

TGA of film samples is necessary.

In general deeper discussion of each topic are necessary.

Conclusions are adequate for the work.

Reviewer 2 Report

In this work, phenolic compounds were extracted from pecan nut cake (PNC) and their application in films prepared from polymer blends composed of gelatin and corn starch was investigated. In addition, the influence of ethanol content and extraction time on the phenolic compounds’ content was investigated, as well as the effects of PNC on the properties of the films. Finally, it was concluded that biodegradable films prepared from a mixture of gelatin-starch and PNC extract enrichment showed excellent mechanical properties and potential as carriers of antioxidant compounds, allowing to propose their use as active packing. When I read this article, I thought it was a very good article with great experimental thoughts and schemes that could easily follow the author's descriptions and explanations. However, a few points deserve attention as the following:

1.       In the abstract section, the abstract should include the purpose, methods, results, and conclusions of the study, not just an accumulation of multiple results.

2.       The introduction section needs some modification. The introduction is not a simple list of individual components, should be clear about your purpose and innovation.

3.       In section 2.4.5, you studied the water vapor permeability of films, and oxygen permeability is also an important way to evaluate films. Please add.

4.       On page 5, line 202. Comparing the TPC content obtained in this paper with the TPC content reported by Galvão Maciel et al. , there is a large variation between the two. You explained that it was due to some factors, such as geographical location, variety, etc. Is there any other literature to prove that PNC of this variety extracts to more content of TPC? Or is there other literature that is consistent with the conclusion you have arrived at?

5.       In Figure 1, please provide a clearer SEM images.

6.       Page 11, line 394. You mentioned that "the addition of natural antioxidants to films can improve the oxidative stability of food product and inhibit the growth of foodborne pathogens on the films". The study of the growth of pathogenic pathogens on the film should study the antibacterial activity of the film, not the antioxidant activity. Please explain.

7.       In section 3.2.10, the biodegradability assessment was given in pictures, can there be data to prove the biodegradation rate of GA and GA with PNC added?

8.       The conclusion described the results of this article, which is good. However, the conclusion should briefly discuss the significance and implications of the study.

Round 2

Reviewer 1 Report

No comments

Reviewer 2 Report

I have no additional comments.